# Nonparametric Bayesian inference on multivariate exponential families

**William Vega-Brown, Marek Doniec, and Nicholas Roy**
Massachusetts Institute of Technology
Cambridge, MA 02139
{wrvb, doniec, nickroy}@csail.mit.edu

## Abstract

We develop a model by choosing the maximum entropy distribution from the set of models satisfying certain smoothness and independence criteria; we show that inference on this model generalizes local kernel estimation to the context of Bayesian inference on stochastic processes. Our model enables Bayesian inference in contexts when standard techniques like Gaussian process inference are too expensive to apply. Exact inference on our model is possible for any likelihood function from the exponential family. Inference is then highly efficient, requiring only $\mathcal{O}\left(\log N\right)$ time and $\mathcal{O}\left(N\right)$ space at run time. We demonstrate our algorithm on several problems and show quantifiable improvement in both speed and performance relative to models based on the Gaussian process.

## 1   Introduction

Many learning problems can be formulated in terms of inference on predictive stochastic models. These models are distributions $p(\boldsymbol{y}|\boldsymbol{x})$ over possible *observation* values $\boldsymbol{y}$ drawn from some observation set $\mathcal{Y}$, conditioned on a known *input* value $x$ from an input set $\mathcal{X}$. The supervised learning problem is then to infer a distribution $p(\boldsymbol{y}|\boldsymbol{x}^*, \mathcal{D})$ over possible observations for some target input $\boldsymbol{x}^*$, given a sequence of $N$ independent observations $\mathcal{D} = \{(\boldsymbol{x}_1, \boldsymbol{y}_1), \ldots, (\boldsymbol{x}_N, \boldsymbol{y}_N)\}$.

It is often convenient to associate latent parameters $\boldsymbol{\theta} \in \boldsymbol{\Theta}$ with each input $\boldsymbol{x}$, where $p(\boldsymbol{y}|\boldsymbol{\theta})$ is a known likelihood function. By inferring a distribution over target parameters $\boldsymbol{\theta}^*$ associated with $\boldsymbol{x}^*$, we can infer a distribution over $\boldsymbol{y}$.

$$p(\boldsymbol{y}|\boldsymbol{x}^*, \mathcal{D}) = \int_{\boldsymbol{\Theta}} \mathrm{d}\boldsymbol{\theta}^*\, p(\boldsymbol{y}|\boldsymbol{\theta}^*)p(\boldsymbol{\theta}^*|\boldsymbol{x}^*, \mathcal{D}) \tag{1}$$

For instance, regression problems can be formulated as the inference of an unknown but deterministic underlying function $\theta(\boldsymbol{x})$ given noisy observations, so that $p(\boldsymbol{y}|\boldsymbol{x}) = \mathcal{N}(\boldsymbol{y}; \theta(\boldsymbol{x}), \sigma^2)$, where $\sigma^2$ is a known noise variance. If we can specify a joint prior over the parameters corresponding to different inputs, we can infer $p(\boldsymbol{\theta}^*|\boldsymbol{x}^*, \mathcal{D})$ using Bayes' rule.

$$p(\boldsymbol{\theta}^*|\boldsymbol{x}^*, \mathcal{D}) \propto \int_{\boldsymbol{\Theta}^N} \left[\prod_{i=1}^{N} \mathrm{d}\boldsymbol{\theta}_i p(\boldsymbol{y}_i|\boldsymbol{\theta}_i)\right] p(\boldsymbol{\theta}_{1:N}, \boldsymbol{\theta}^*|\boldsymbol{x}^*, \boldsymbol{x}_{1:N}) \tag{2}$$

The notation $\boldsymbol{x}_{1:N}$ indicates the sample inputs $\{\boldsymbol{x}_1, \ldots, \boldsymbol{x}_N\}$; this model is depicted graphically in figure 1a. Although the choice of likelihood is often straightforward, specifying a prior can be more difficult. Ideally, we want a prior which is expressive, in the sense that it can accurately capture all prior knowledge, and which permits efficient and accurate inference.

A powerful motivating example for specifying problems in terms of generative models is the Gaussian process [1], which specifies the prior $p(\boldsymbol{\theta}_{1:N}|\boldsymbol{x}_{1:N})$ as a multivariate Gaussian with a covariance parameterized by $\boldsymbol{x}_{1:N}$. This prior can express complex and subtle relationships between inputs and

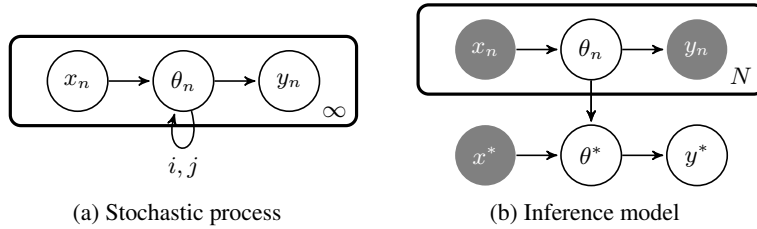

(a) Stochastic process          (b) Inference model

Figure 1: *Figure 1a models any stochastic process with fully connected latent parameters. Figure 1b is our approximate model, used for inference; we assume that the latent parameters are independent given the target parameters. Shaded nodes are observed.*

observations, and permits efficient exact inference for a Gaussian likelihood with known variance. Extensions exist to perform approximate inference with other likelihood functions [2, 3, 4, 5].

However, the assumptions of the Gaussian process are not the only set of reasonable assumptions, and are not always appropriate. Very large datasets require complex sparsification techniques to be computationally tractable [6]. Likelihood functions with many coupled parameters, such as the parameters of a categorical distribution or of the covariance matrix of a multivariate Gaussian, require the introduction of large numbers of latent variables which must be inferred approximately. As an example, the generalized Wishart process developed by Wilson and Ghahramani [7] provides a distribution over covariance matrices using a sum of Gaussian processes. Inference of the the posterior distribution over the covariance can only be performed approximately: no exact inference procedure is known.

Historically, an alternative approach to estimation has been to use local kernel estimation techniques [8, 9, 10], which are often developed from a weighted parameter likelihood of the form $p(\boldsymbol{\theta}|\mathcal{D}) = \prod_i p(\boldsymbol{y}_i|\boldsymbol{\theta})^{w_i}$. Algorithms for determining the maximum likelihood parameters of such a model are easy to implement and very fast in practice; various techniques, such as dual trees [11] or the improved fast Gauss transform [12] allow the computation of kernel estimates in logarithmic or constant time. This choice of model is often principally motivated by the computational convenience of resulting algorithms. However, it is not clear how to perform Bayesian inference on such models. Most instantiations instead return a point estimate of the parameters.

In this paper, we bridge the gap between the local kernel estimators and Bayesian inference. Rather than perform approximate inference on an exact generative model, we formulate a simplified model for which we can efficiently perform exact inference. Our simplification is to choose the maximum entropy distribution from the set of models satisfying certain smoothness and independence criteria. We then show that for any likelihood function in the exponential family, our process model has a conjugate prior, which permits us to perform Bayesian inference in closed form. This motivates many of the local kernel estimators from a Bayesian perspective, and generalizes them to new problem domains. We demonstrate the usefulness of this model on multidimensional regression problems with coupled observations and input-dependent noise, a setting which is difficult to model using Gaussian process inference.

## 2 The kernel process model

Given a likelihood function, a generative model can be specified by a prior $p(\boldsymbol{\theta}_{1:N}, \boldsymbol{\theta}^*|\boldsymbol{x}^*, \boldsymbol{x}_{1:N})$. For almost all combinations of prior and likelihood, inference is analytically intractable. We relax the requirement that the model be generative, and instead require only that the prior be well-formed for a given query $\boldsymbol{x}^*$. To facilitate inference, we make the strong assumption that the latent training parameters $\boldsymbol{\theta}_{1:N}$ are conditionally independent given the target parameters $\boldsymbol{\theta}^*$.

$$p(\boldsymbol{\theta}_{1:N}, \boldsymbol{\theta}^*|\boldsymbol{x}_{1:N}, \boldsymbol{x}^*) = \left[ \prod_{i=1}^{N} p(\boldsymbol{\theta}_i|\boldsymbol{\theta}^*, \boldsymbol{x}_i, \boldsymbol{x}^*) \right] p(\boldsymbol{\theta}^*|\boldsymbol{x}^*) \qquad (3)$$

This restricted structure is depicted graphically in figure 1b. Essentially, we assume that interactions between latent parameters are unimportant relative to interactions between the latent and target parameters; this will allow us to build models based on exponential family likelihood functions which will permit exact inference. Note that models with this structure will not correspond exactly to probabilistic generative models; the prior distribution over the latent parameters associated with the training inputs varies depending on the target input. Instead of approximating inference on our

model, we make our approximation at the stage of model selection; in doing so, we enable fast, exact inference. Note that the class of models with the structure of equation (3) is quite rich, and as we demonstrate in section 3.2, performs well on many problems. We discuss in section 4 the ramifications of this assumption and when it is appropriate.

This assumption is closely related to known techniques for sparsifying Gaussian process inference. Quiñonero-Candela and Rasmussen [6] provide a summary of many sparsification techniques, and describe which correspond to generative models. One of the most successful sparsification techniques, the fully independent training conditional approximation of Snelson [13], assumes all training examples are independent given a specified set of inducing inputs. Our assumption extends this to the case of a single inducing input equal to the target input.

Note that by marginalizing the parameters $\boldsymbol{\theta}_{1:N}$, we can directly relate the observations $\boldsymbol{y}_{1:N}$ to the target parameters $\boldsymbol{\theta}^*$. Combining equations (2) and (3),

$$p(\boldsymbol{\theta}^*|\boldsymbol{x}^*, \mathcal{D}) \propto \left[ \prod_{i=1}^{N} \int_{\Theta} \mathrm{d}\boldsymbol{\theta}_i p(\boldsymbol{y}_i|\boldsymbol{\theta}_i) p(\boldsymbol{\theta}_i|\boldsymbol{\theta}^*, \boldsymbol{x}_i, \boldsymbol{x}^*) \right] p(\boldsymbol{\theta}^*|\boldsymbol{x}^*) \tag{4}$$

and marginalizing the latent parameters $\boldsymbol{\theta}_{1:N}$, we observe that the posterior factors into a product over likelihoods $p(\boldsymbol{y}_i|\boldsymbol{\theta}^*, \boldsymbol{x}, \boldsymbol{x}^*)$ and a prior over $\boldsymbol{\theta}^*$.

$$= \left[ \prod_{i=1}^{N} p(\boldsymbol{y}_i|\boldsymbol{\theta}^*, \boldsymbol{x}_i, \boldsymbol{x}^*) \right] p(\boldsymbol{\theta}^*|\boldsymbol{x}^*) \tag{5}$$

Note that we can equivalently specify either $p(\boldsymbol{\theta}|\boldsymbol{\theta}^*, \boldsymbol{x}, \boldsymbol{x}^*)$ or $p(\boldsymbol{y}|\boldsymbol{\theta}^*, \boldsymbol{x}, \boldsymbol{x}^*)$, without loss of generality. In other words, we can equivalently describe the interaction between input variables either in the likelihood function or in the prior.

## 2.1 The extended likelihood function

By construction, we know the distribution $p(\boldsymbol{y}_i|\boldsymbol{\theta}_i)$. After making the transformation to equation (5), much of the problem of model specification has shifted to specifying the distribution $p(\boldsymbol{y}_i|\boldsymbol{\theta}^*, \boldsymbol{x}_i, \boldsymbol{x}^*)$. We call this distribution the extended likelihood distribution. Intuitively, these distributions should be related; if $\boldsymbol{x}^* = \boldsymbol{x}_i$, then we expect $\boldsymbol{\theta}_i = \boldsymbol{\theta}^*$ and $p(\boldsymbol{y}_i|\boldsymbol{\theta}^*, \boldsymbol{x}_i, \boldsymbol{x}^*) = p(\boldsymbol{y}_i|\boldsymbol{\theta}_i)$. We therefore define the *extended* likelihood function in terms of the known likelihood $p(\boldsymbol{y}_i|\boldsymbol{\theta}_i)$.

Typically, we prefer smooth models: we expect similar inputs to lead to a similar distribution over outputs. In the absence of a smoothness constraint, any inference method can perform arbitrarily poorly [14]. However, the notion of smoothness is not well-defined in the context of probability distributions. Denote $g(\boldsymbol{y}_i) = p(\boldsymbol{y}_i|\boldsymbol{\theta}^*, \boldsymbol{x}_i, \boldsymbol{x}^*)$, and $f(\boldsymbol{y}_i) = p(\boldsymbol{y}_i|\boldsymbol{\theta}_i)$. We can formalize a *smooth model* as one in which the information divergence of the likelihood distribution $f$ from the extended likelihood distribution $g$ is *bounded* by some function $\rho : \mathcal{X} \times \mathcal{X} \to \mathbb{R}^+$.

$$D_{\mathrm{KL}}\left(g\|f\right) \leq \rho(\boldsymbol{x}^*, \boldsymbol{x}_i) \tag{6}$$

Since the divergence is a premetric, $\rho(\cdot, \cdot)$ must also satisfy the properties of a premetric: $\rho(\boldsymbol{x}, \boldsymbol{x}) = 0 \,\forall \boldsymbol{x}$ and $\rho(\boldsymbol{x}_1, \boldsymbol{x}_2) \geq 0 \,\forall \boldsymbol{x}_1, \boldsymbol{x}_2$. For example, if $\mathcal{X} = \mathbb{R}^n$, we may draw an analogy to Lipschitz continuity and choose $\rho(\boldsymbol{x}_1, \boldsymbol{x}_2) = K\|\boldsymbol{x}_1 - \boldsymbol{x}_2\|$, with $K$ a positive constant. The class of models with bounded divergence has the property that $g \to f$ as $\boldsymbol{x}' \to \boldsymbol{x}$, and it does so smoothly provided $\rho(\cdot, \cdot)$ is smooth. Note that this bound is a constraint on possible $g$, not an objective to be minimized; in particular, we do not minimize the divergence between $g$ and $f$ to develop an approximation, as is common in the approximate inference literature. Note also that this constraint has a straightforward information-theoretic interpretation; $\rho(\boldsymbol{x}_1, \boldsymbol{x}_2)$ is a bound on the amount of information we would lose if we were to assume an observation $\boldsymbol{y}_1$ were taken at $\boldsymbol{x}_2$ instead of at $\boldsymbol{x}_1$.

The assumptions of equations (3) and (6) define a class of models for a given likelihood function, but are insufficient for specifying a well-defined prior. We therefore use the principle of maximum entropy and choose the maximum entropy distribution from among that class. In our attached supporting material, we prove the following.

**Theorem 1** *The maximum entropy distribution $g$ satisfying $D_{\mathrm{KL}}\left(g\|f\right) \leq \rho(\boldsymbol{x}^*, \boldsymbol{x})$ has the form*

$$g(\boldsymbol{y}) \propto f(\boldsymbol{y})^{k(\boldsymbol{x}^*, \boldsymbol{x})} \tag{7}$$

*where $k : \mathcal{X} \times \mathcal{X} \to [0, 1]$ is a kernel function which can be uniquely determined from $\rho(\cdot, \cdot)$ and $f(\cdot)$.*

There is an equivalence relationship between functions $k(\cdot,\cdot)$ and $\rho(\cdot,\cdot)$; as either is uniquely determined by the other, it may more convenient to select a kernel function than a smoothness bound, and doing so implies no loss in generality or correctness. Note it is neither necessary nor sufficient that the kernel function $k(\cdot,\cdot)$ be positive definite. It is necessary only that $k(\boldsymbol{x},\boldsymbol{x}) = 1 \forall \boldsymbol{x}$ and that $k(\boldsymbol{x},\boldsymbol{x}') \in [0,1] \forall \boldsymbol{x},\boldsymbol{x}'$. This includes the possibility of asymmetric kernel functions. We discuss in the attached supporting material the mapping between valid kernel functions $k(\cdot,\cdot)$ and bounding functions $\rho(\cdot,\cdot)$.

It follows from equation (7) that the maximum entropy distribution satisfying a bound of $\rho(\boldsymbol{x},\boldsymbol{x}^*)$ on the divergence of the observation distribution $p(\boldsymbol{y}|\boldsymbol{\theta}^*,\boldsymbol{x},\boldsymbol{x}^*)$ from the known distribution $p(\boldsymbol{y}|\boldsymbol{\theta},\boldsymbol{x},\boldsymbol{x}^*) = p(\boldsymbol{y}|\boldsymbol{\theta})$ is

$$p(\boldsymbol{y}|\boldsymbol{\theta}^*,\boldsymbol{x},\boldsymbol{x}^*) \propto p(\boldsymbol{y}|\boldsymbol{\theta})^{k(\boldsymbol{x},\boldsymbol{x}^*)}. \tag{8}$$

By combining equations (5) and (6), we can fully specify a stochastic model with a likelihood $p(\boldsymbol{y}|\boldsymbol{\theta})$, a pointwise marginal prior $p(\boldsymbol{\theta}|\boldsymbol{x})$, and a kernel function $k : \mathcal{X} \times \mathcal{X} \to [0,1]$. To perform inference, we must evaluate

$$p(\boldsymbol{\theta}|\boldsymbol{x},\mathcal{D}) \propto \prod_{i=1}^{N} p(\boldsymbol{y}_i|\boldsymbol{\theta})^{k(\boldsymbol{x},\boldsymbol{x}_i)} p(\boldsymbol{\theta}|\boldsymbol{x}) \tag{9}$$

This can be done in closed form if we can normalize the terms on the right side of the equality.

In certain limiting cases with uninformative priors, our model can be reduced to known frequentist estimators. For instance, if we employ an uninformative prior $p(\boldsymbol{\theta}|\boldsymbol{x}) \propto 1$ and choose the maximum-likelihood target parameters $\hat{\boldsymbol{\theta}}^* = \arg\max p(\boldsymbol{\theta}^*|\boldsymbol{x}^*,\mathcal{D})$, we recover the weighted maximum-likelihood estimator, detailed by Wang [15]. If the function $k(\boldsymbol{x},\boldsymbol{x}')$ is local, in the sense that it goes to zero if the distance $\|\boldsymbol{x} - \boldsymbol{x}'\|$ is large, then choosing maximum likelihood parameter estimates for an uninformative prior gives the locally weighted maximum-likelihood estimator, described in the context of regression by Cleveland [16] and for generalized linear models by Tibshirani and Hastie [10]. However, our result is derived from a Bayesian interpretation of statistics, and we infer a full distribution over the parameters; we are not limited to a point estimate. The distinction is of both academic and practical interest; in addition to providing insight into to the meaning of the weighting function and the validity of the inferred parameters, by inferring a posterior distribution we provide a principled way to reason about our knowledge and to insert prior knowledge of the underlying process.

## 2.2 Kernel inference on the exponential family

Equation (8) is particularly useful if we choose our likelihood model $p(\boldsymbol{y}|\boldsymbol{\theta})$ from the exponential family.

$$p(\boldsymbol{y}|\boldsymbol{\theta}) = h(\boldsymbol{y})\exp\left(\boldsymbol{\theta}^\top \boldsymbol{T}(\boldsymbol{y}) - A(\boldsymbol{\theta})\right) \tag{10}$$

A member of an exponential family remains in the same family when raised to the power of $k(\boldsymbol{x},\boldsymbol{x}_i)$. Because every exponential family has a conjugate prior, we may choose our point-wise prior $p(\boldsymbol{\theta}^*|\boldsymbol{x}^*)$ to be conjugate to our chosen likelihood. We denote this conjugate prior $p_\pi(\boldsymbol{\chi},\nu)$, where $\boldsymbol{\chi}$ and $\nu$ are hyperparameters.

$$p(\boldsymbol{\theta}|\boldsymbol{x}^*) = p_\pi(\boldsymbol{\chi}(\boldsymbol{x}^*),\nu(\boldsymbol{x}^*)) = f(\boldsymbol{\chi}(\boldsymbol{x}^*),\nu(\boldsymbol{x}^*))\exp\left(\boldsymbol{\theta} \cdot \boldsymbol{\chi}(\boldsymbol{x}^*) - \nu(\boldsymbol{x}^*)A(\boldsymbol{\theta})\right) \tag{11}$$

Therefore, our posterior as defined by equation (9) may be evaluated in closed form.

$$p(\boldsymbol{\theta}^*|\boldsymbol{x}^*,\mathcal{D}) = p_\pi(\sum_{i=1}^{N} k(\boldsymbol{x}^*,\boldsymbol{x}_i)\boldsymbol{T}(\boldsymbol{y}_i) + \boldsymbol{\chi}(\boldsymbol{x}^*), \sum_{i=1}^{N} k(\boldsymbol{x}^*,\boldsymbol{x}_i) + \nu(\boldsymbol{x}^*)) \tag{12}$$

The prior predictive distribution $p(\boldsymbol{y}|\boldsymbol{x})$ is given by

$$p(\boldsymbol{y}|\boldsymbol{x}) = \int p(\boldsymbol{y}|\boldsymbol{\theta})p_\pi(\boldsymbol{\theta}|\boldsymbol{\chi}(\boldsymbol{x}^*),\nu(\boldsymbol{x}^*)) \tag{13}$$

$$= h(\boldsymbol{y})\frac{f(\boldsymbol{\chi}(\boldsymbol{x}^*),\nu(\boldsymbol{x}^*))}{f(\boldsymbol{\chi}(\boldsymbol{x}^*) + \boldsymbol{T}(\boldsymbol{y}),\nu(\boldsymbol{x}^*) + 1)} \tag{14}$$

and the posterior predictive distribution is

$$p(\boldsymbol{y}|\boldsymbol{x}^*, \mathcal{D}) = h(\boldsymbol{y}) \frac{f(\sum_{i=1}^N k(\boldsymbol{x}^*, \boldsymbol{x}_i)\boldsymbol{T}(\boldsymbol{y}_i) + \boldsymbol{\chi}(\boldsymbol{x}^*), \sum_{i=1}^N k(\boldsymbol{x}^*, \boldsymbol{x}_i) + \nu(\boldsymbol{x}^*))}{f(\sum_{i=1}^N k(\boldsymbol{x}^*, \boldsymbol{x}_i)\boldsymbol{T}(\boldsymbol{y}_i) + \boldsymbol{\chi}(\boldsymbol{x}^*) + \boldsymbol{T}(\boldsymbol{y}), \sum_{i=1}^N k(\boldsymbol{x}^*, \boldsymbol{x}_i) + \nu(\boldsymbol{x}^*) + 1)}$$
(15)

This is a general formulation of the posterior distribution over the parameters of any likelihood model belonging to the exponential family. Note that given a function $k(\boldsymbol{x}^*, \boldsymbol{x})$, we may evaluate this posterior without sampling, in time linear in the number of samples. Moreover, for several choices of kernels the relevant sums can be evaluates in sub-linear time; a sum over squared exponential kernels, for instance, can be evaluated in logarithmic time.

## 3  Local inference for multivariate Gaussian

We now discuss in detail the application of equation (12) to the case of a multivariate Gaussian likelihood model with unknown mean $\boldsymbol{\mu}$ and unknown covariance $\boldsymbol{\Sigma}$.

$$p(\boldsymbol{y}|\boldsymbol{\mu}, \boldsymbol{\Sigma}) = \mathcal{N}(\boldsymbol{y}; \boldsymbol{\mu}, \boldsymbol{\Sigma})$$
(16)

We present the conjugate prior, posterior, and predictive distributions without derivation; see [17], for example, for a derivation. The conjugate prior for a multivariate Gaussian with unknown mean and covariance is the normal-inverse Wishart distribution, with hyperparameter functions $\boldsymbol{\mu}_0(\boldsymbol{x}^*)$, $\boldsymbol{\Psi}(\boldsymbol{x}^*)$, $\nu(\boldsymbol{x}^*)$, and $\lambda(\boldsymbol{x}^*)$.

$$p(\boldsymbol{\mu}, \boldsymbol{\Sigma}|\boldsymbol{x}^*) = \mathcal{N}\left(\boldsymbol{\mu}; \boldsymbol{\mu}_0(\boldsymbol{x}^*), \frac{\boldsymbol{\Sigma}}{\lambda(\boldsymbol{x}^*)}\right) \times \mathcal{W}^{-1}(\boldsymbol{\Sigma}; \boldsymbol{\Psi}(\boldsymbol{x}^*), \nu(\boldsymbol{x}^*))$$
(17)

The hyperparameter functions have intuitive interpretations; $\boldsymbol{\mu}_0(\boldsymbol{x}^*)$ is our initial belief of the mean function, while $\lambda(\boldsymbol{x}^*)$ is our confidence in that belief, with $\lambda(\boldsymbol{x}^*) = 0$ indicating no confidence in the region near $\boldsymbol{x}^*$, and $\lambda(\boldsymbol{x}^*) \to \infty$ indicating a state of perfect knowledge. Likewise, $\boldsymbol{\Psi}(\boldsymbol{x}^*)$ indicates the expected covariance, and $\nu(\boldsymbol{x}^*)$ represents the confidence in that estimate, much like $\lambda$. Given a dataset $\mathcal{D}$, we can compute a posterior over the mean and covariance, represented by updated parameters $\boldsymbol{\mu}_0'(\boldsymbol{x}^*)$, $\boldsymbol{\Psi}'(\boldsymbol{x}^*)$, $\lambda'(\boldsymbol{x}^*)$, and $\nu'(\boldsymbol{x}^*)$.

$$\lambda'(\boldsymbol{x}^*) = \lambda(\boldsymbol{x}^*) + \overline{k}(\boldsymbol{x}^*) \qquad \nu'(\boldsymbol{x}^*) = \nu(\boldsymbol{x}^*) + \overline{k}(\boldsymbol{x}^*)$$

$$\boldsymbol{\mu}_0'(\boldsymbol{x}^*) = \frac{\lambda(\boldsymbol{x}^*)\boldsymbol{\mu}_0(\boldsymbol{x}^*) + \overline{\boldsymbol{y}}}{\lambda(\boldsymbol{x}^*) + \overline{k}(\boldsymbol{x}^*)}$$
(18)

$$\boldsymbol{\Psi}'(\boldsymbol{x}^*) = \boldsymbol{\Psi}(\boldsymbol{x}^*) + \overline{\boldsymbol{S}}(\boldsymbol{x}^*) + \frac{\lambda(\boldsymbol{x}^*)\overline{k}(\boldsymbol{x}^*)}{\lambda(\boldsymbol{x}^*) + \overline{k}(\boldsymbol{x}^*)}\overline{\boldsymbol{E}}(\boldsymbol{x}^*)$$

where

$$\overline{k}(\boldsymbol{x}^*) = \sum_{i=1}^N k(\boldsymbol{x}^*, \boldsymbol{x}_i) \qquad \overline{\boldsymbol{y}}(\boldsymbol{x}^*) = \frac{1}{\overline{k}(\boldsymbol{x}^*)}\sum_{i=1}^N k(\boldsymbol{x}^*, \boldsymbol{x}_i)\boldsymbol{y}_i$$

$$\overline{\boldsymbol{S}}(\boldsymbol{x}^*) = \sum_{i=1}^N k(\boldsymbol{x}^*, \boldsymbol{x}_i)\left(\boldsymbol{y}_i - \overline{\boldsymbol{y}}(\boldsymbol{x}^*)\right)\left(\boldsymbol{y}_i - \overline{\boldsymbol{y}}(\boldsymbol{x}^*)\right)^\top$$
(19)

$$\overline{\boldsymbol{E}}(\boldsymbol{x}^*) = \left(\overline{\boldsymbol{y}}(\boldsymbol{x}^*) - \boldsymbol{\mu}_0(\boldsymbol{x}^*)\right)\left(\overline{\boldsymbol{y}}(\boldsymbol{x}^*) - \boldsymbol{\mu}_0(\boldsymbol{x}^*)\right)^\top$$

The resulting posterior predictive distribution is a multivariate Student-$t$ distribution.

$$p(\boldsymbol{y}|\boldsymbol{x}^*) = t_{\nu'(\boldsymbol{x}^*)}\left(\boldsymbol{\mu}_0'(\boldsymbol{x}^*), \frac{\lambda'(\boldsymbol{x}^*) + 1}{\lambda'(\boldsymbol{x}^*)\nu'(\boldsymbol{x}^*)}\boldsymbol{\Psi}'(\boldsymbol{x}^*)\right)$$
(20)

### 3.1  Special cases

Two special cases of the multivariate Gaussian are worth mentioning. First, a fixed, known covariance $\Sigma(\boldsymbol{x}^*)$ can be described by the hyperparameters $\Psi(\boldsymbol{x}^*) = \lim_{\nu \to \infty} \frac{\Sigma(\boldsymbol{x}^*)}{\nu}$. The resulting posterior distribution is then

$$p(\mu|\boldsymbol{x}^*, \mathcal{D}) = \mathcal{N}\left(\boldsymbol{\mu}_0', \frac{1}{\lambda'(\boldsymbol{x}^*)}\Sigma(\boldsymbol{x}^*)\right)$$
(21)

with predictive distribution

$$p(\mu|\boldsymbol{x}^*, \mathcal{D}) = \mathcal{N}\left(\boldsymbol{\mu}_0', \frac{1 + \lambda'(\boldsymbol{x}^*)}{\lambda'(\boldsymbol{x}^*)}\boldsymbol{\Sigma}(\boldsymbol{x}^*)\right) \tag{22}$$

In the limit as $\lambda$ goes to 0, when the prior is uninformative, the mean and mode of the predictive distribution approaches the Nadaraya-Watson [8, 9] estimate.

$$\mu_{NW}(\boldsymbol{x}^*) = \frac{\sum_{i=1}^{N} k(\boldsymbol{x}^*, \boldsymbol{x}_i)y_i}{\sum_{i=1}^{N} k(\boldsymbol{x}^*, \boldsymbol{x}_i)} \tag{23}$$

The complementary case of known mean $\boldsymbol{\mu}(\boldsymbol{x}^*)$ and unknown covariance $\boldsymbol{\Sigma}(\boldsymbol{x}^*)$ is described by the limit $\lambda \to \infty$. In this case, the posterior distribution is

$$p(\boldsymbol{\Sigma}|\boldsymbol{x}^*, \mathcal{D}) = \mathcal{W}^{-1}\left(\boldsymbol{\Psi}(\boldsymbol{x}^*) + \sum_{i=1}^{N} k_i(\boldsymbol{y}_i - \boldsymbol{\mu}(\boldsymbol{x}^*))(\boldsymbol{y}_i - \boldsymbol{\mu}(\boldsymbol{x}^*))^\top, \lambda(\boldsymbol{x}^*) + \sum_{i=1}^{N} k_i\right) \tag{24}$$

with predictive distribution

$$p(\boldsymbol{y}|\boldsymbol{x}^*) = t_{\nu'(\boldsymbol{x}^*)}\left(\boldsymbol{\mu}(\boldsymbol{x}^*), \frac{1}{\nu'(\boldsymbol{x}^*)}\boldsymbol{\Psi}(\boldsymbol{x}^*) + \sum_{i=1}^{N} k_i(\boldsymbol{y}_i - \boldsymbol{\mu}(\boldsymbol{x}^*))(\boldsymbol{y}_i - \boldsymbol{\mu}(\boldsymbol{x}^*))^\top\right) \tag{25}$$

In the limit as $\nu$ goes to 0, the maximum likelihood covariance estimate is

$$\boldsymbol{\Sigma}_{\mathrm{ML}}(\boldsymbol{x}^*) = \sum_{i=1}^{N} k_i(\boldsymbol{y}_i - \boldsymbol{\mu}(\boldsymbol{x}^*))(\boldsymbol{y}_i - \boldsymbol{\mu}(\boldsymbol{x}^*))^\top \tag{26}$$

which is precisely the result of our prior work [18, 19]. In both cases, our method yields distributions over parameters, rather than point estimates; moreover, the use of Bayesian inference naturally handles the case of limited or no available samples.

## 3.2 Experimental results

We evaluate our approach on several regression problems, and compare the results with alternative nonparametric Bayesian models. In all experiments, we use the squared-exponential kernel $k(\boldsymbol{y}, \boldsymbol{y}') = \exp(\frac{c}{2}\|\boldsymbol{y} - \boldsymbol{y}'\|^2)$. This function meets both the requirements of our algorithm and is positive-definite and thus a suitable covariance function for models based on the Gaussian process. We set the kernel scale $c$ by maximum likelihood for each model.

We compare our approach to covariance prediction to the generalized Wishart process (GWP) of [7]. First, we sample a synthetic dataset; the output is a two-dimensional observation set $\mathcal{Y} = \mathbb{R}^2$, where samples are drawn from a zero-mean normal distribution with a covariance that rotates over time.

$$\boldsymbol{\Sigma}(t) = \begin{pmatrix} \cos(t) & -\sin(t) \\ \sin(t) & \cos(t) \end{pmatrix} \begin{pmatrix} 4 & 0 \\ 0 & 10 \end{pmatrix} \begin{pmatrix} \cos(t) & -\sin(t) \\ \sin(t) & \cos(t) \end{pmatrix}^\top \tag{27}$$

Second, we predict the covariances of the returns on two currency exchanges—the Euro to US dollar, and the Japanese yen to US dollar—over the past four years. Following Wilson and Ghahramani, we define a return as $\log(\frac{P_{t+1}}{P_t})$, where $P_t$ is the exchange rate on day $t$. Illustrative results are provided in figure 2. To compare these results quantitatively, one natural measure is the mean of the logarithm of the likelihood of the predicted model given the data.

$$\mathrm{MLL} = \frac{1}{N} \sum_{i=1}^{N} -\frac{1}{2}(\boldsymbol{y}_i^\top \hat{\boldsymbol{\Sigma}}_i^{-1} \boldsymbol{y}_i + \log \det \hat{\boldsymbol{\Sigma}}_i) \tag{28}$$

Here, $\hat{\boldsymbol{\Sigma}}_i$ is the maximum likelihood covariance predicted for the $i^{\mathrm{th}}$ sample.

In addition to how well our model describes the available data, we may also be interested in how accurately we recover the distribution used to generate the data. This is a measure of how closely the inferred ellipses in figure 2 approximate the true covariance ellipses. One measure of the quality of the inferred distribution is the KL divergence of the inferred distribution from the true distribution

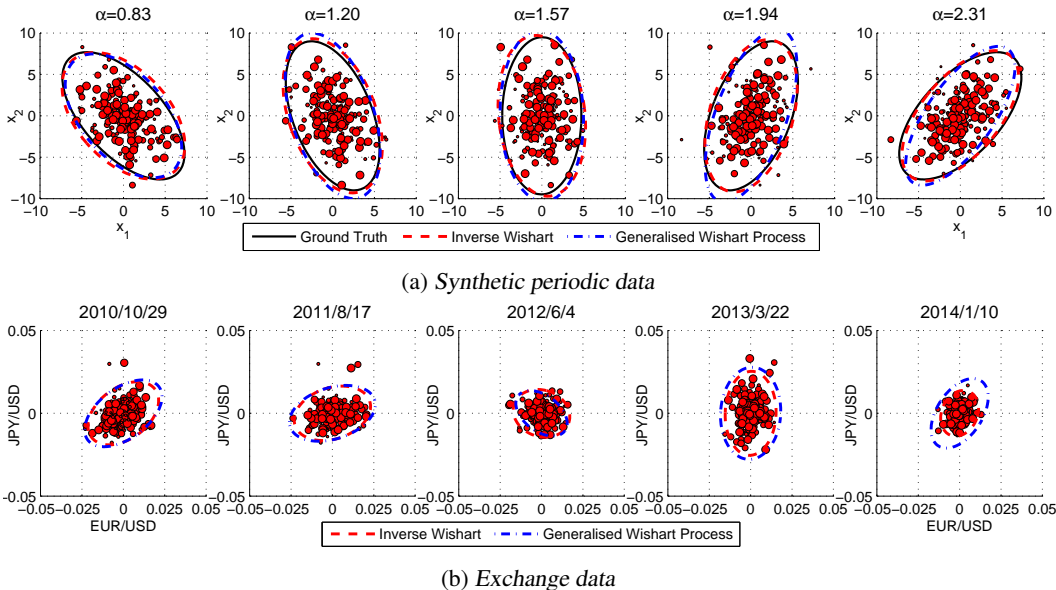

(a) *Synthetic periodic data*

(b) *Exchange data*

Figure 2: *Comparison of covariances predicted by our kernel inverse Wishart process and the generalized Wishart process for the problems described in section 3.2. The true covariance used to generate data is provided for comparison. The samples used are plotted so that the area of the circle is proportional to the weight assigned by the kernel. The kernel inverse Wishart process outperforms the generalized Wishart process, both in terms of the likelihood of the training data, and in terms of the divergence of the inferred distribution from the true distribution.*

used to generate the data. Note we cannot evaluate this quantity on the exchange dataset, as we do not know the true distribution. We present both the mean likelihood and the KL divergence of both algorithms, along with running times, in table 1.

By both metrics, our algorithm outperforms the GWP by a significant margin; the running time advantage of kernel estimation over the GWP is even more dramatic. It is important to note that running times are difficult to compare, as they depend heavily on implementation and hardware details; the numbers reported should be considered qualitatively. Both algorithms were implemented in the MATLAB programming language, with the likelihood functions for the GWP implemented in heavily optimized C code in an effort to ensure a fair competition. Despite this, the GWP took over a thousand times longer than our method to generate predictions.

|  |  | $t_{tr}$ (s) | $t_{ev}$ (ms) | MLL | $D_{\mathrm{KL}}\left(\hat{p}\|p\right)$ |
|---|---|---|---|---|---|
| Periodic | kNIW | **0.022** | **0.003** | **-10.43** | **0.0138** |
|  | GWP | 7.08 | 0.135 | -19.79 | 0.0248 |
| Exchange | kNIW | **0.520** | **0.020** | **7.73** | — |
|  | GWP | 15.7 | 1.708 | 7.56 | — |

Table 1: *Comparison of the performance of two models of covariance prediction, based on time required to make predictions at evaluation, the mean log likelihood and the KL divergence between the predicted covariance and the ground truth covariance.*

We next evaluate our approach on heteroscedastic regression problems. First, we generate 100 samples from the distribution described by Yuan and Wahba [20], which has mean $\mu(\boldsymbol{x}^*) = 2\exp(-30(\boldsymbol{x}^* - 0.25)^2) + \sin(\pi(\boldsymbol{x}^*)^2)$ and variance $\sigma^2(\boldsymbol{x}^*) = \exp(2 * \sin(2\pi\boldsymbol{x}^*))$. Second, we test on the motorcycle dataset of Silverman et al. [21]. We compare our approach to a variety of Gaussian process based regression algorithms, including a standard homoscedastic Gaussian process, the variational heteroscedastic Gaussian process of Lázaro-Gredilla and Titsias [4], and the maximum likelihood heteroscedastic Gaussian process of Quadrianto et al. [22]. All algorithms are implemented in MATLAB, using the authors' own code. Running times are presented with the same caveat as in the previous experiments, and a similar conclusion holds: our method provides results which are as good or better than methods based upon the Gaussian process, and does so in a fraction of the time. Figure 3 illustrates the predictions made by our method on the heteroscedastic motor-

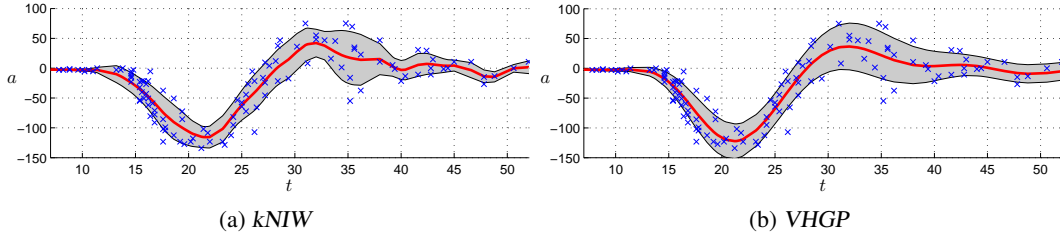

(a) *kNIW*          (b) *VHGP*

Figure 3: *Comparison of the distributions inferred using the kernel normal inverse Wishart process and the variational heteroscedastic Gaussian process to model Silverman's motorcycle dataset. Both models capture the time-varying nature of the measurement noise; as is typical, the kernel model is much less smooth and has more local structure than the Gaussian process model. Both models perform well according to most metrics, but the kernel model can be computed in a fraction of the time.*

cycle dataset of Silverman. For reference, we provide the distribution generated by the variational heteroscedastic Gaussian process.

|  |  | $t_{tr}$ (s) | $t_{ev}$ (ms) | NMSE | MLL |
|---|---|---|---|---|---|
| Motorcycle | kNIW | **0.124** | **2.95** | **0.2** | -4.04 |
|  | GP | 0.52 | 3.52 | 0.202 | -4.51 |
|  | VHGP | 3.12 | 7.53 | 0.202 | -4.07 |
|  | MLHGP | 2.39 | 5.83 | 0.204 | **-4.03** |
| Periodic | kNIW | **0.68** | **7.94** | **0.0708** | -2.07 |
|  | GP | 3.41 | 22 | 0.0822 | -2.56 |
|  | VHGP | 26.4 | 54.4 | 0.0827 | **-1.85** |
|  | MLHGP | 38.3 | 29.1 | 0.0827 | -2.38 |

Table 2: *Comparison of the performance of various models of heteroscedastic processes, based on time required to train, time required to make predictions at evaluation, the normalized mean squared error, and the mean log likelihood. Note how the normal-inverse Wishart process obtains performance as good or better than the other algorithms in a fraction of the time.*

## 4   Discussion

We have presented a family of stochastic models which permit exact inference for any likelihood function from the exponential family. Algorithms for performing inference on this model include many local kernel estimators, and extend them to probabilistic contexts. We showed the instantiation of our model for a multivariate Gaussian likelihood; due to lack of space, we do not present others, but the approach is easily extended to tasks like classification and counting. The models we develop are built on a strong assumption of independence; this assumption is critical to enabling efficient exact inference. We now explore the costs of this assumption, and when it is inappropriate.

First, while the kernel function in our model does not need to be positive definite—or even symmetric—we lose an important degree of flexibility relative to the covariance functions employed in a Gaussian process. Covariance functions can express a number of complex concepts, such as a prior over functions with a specified additive or hierarchical structure [23]; these concepts cannot be easily formulated in terms of smoothness. Second, by neglecting the relationships between latent parameters, we lose the ability to extrapolate trends in the data, meaning that in places where data is sparse we cannot expect good performance. Thus, for a problem like time series forecasting, our approach will likely be unsuccessful. Our approach is suitable in situations where we are likely to see similar inputs many times, which is often the case. Moreover, regardless of the family of models used, extrapolation to regions of sparse data can perform very poorly if the prior does not model the true process well. Our approach is particularly effective when data is readily available, but computation is expensive; the gains in efficiency due to an independence assumption allow us to scale to larger much larger datasets, improving predictive performance with less design effort.

**Acknowledgements**

This research was funded by the Office of Naval Research under contracts N00014-09-1-1052 and N00014-10-1-0936. The support of Behzad Kamgar-Parsi and Tom McKenna is gratefully acknowledged.

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
