[Supplementary Material]

# Nonparametric Bayesian inference on multivariate exponential families
### Supplementary material

**William Vega-Brown, Marek Doniec, and Nicholas Roy**
Massachusetts Institute of Technology
Cambridge, MA 02139
{wrvb, doniec, nickroy}@csail.mit.edu

## 1  Proof of the form of the extended likelihood

Consider the variational optimization problem

$$\underset{p}{\text{maximize}} \qquad \mathcal{F}[p] \tag{1}$$

$$\text{subject to} \qquad \mathcal{G}_i[p] \leq 0 \qquad \forall i \in [1, m] \tag{2}$$

$$\mathcal{H}_j[p] = 0 \forall j \in [1, n] \tag{3}$$

where $\mathcal{F}, \mathcal{G}_i, \mathcal{H}_j$ are functionals $l_1(\mathcal{X}) \to \mathbb{R}$ and $l_1(\mathcal{X})$ is the space of integrable functions over a measurable space $\mathcal{X}$. A solution $p^* \in l_1(\mathcal{X})$ must satisfy the Karush-Kuhn-Tucker conditions.

$$\delta\mathcal{F}[p^*] = \sum_{i=1}^{m} \mu_i \delta\mathcal{G}_i[p^*] + \sum_{j=1}^{n} \lambda_j \delta\mathcal{H}_j[p^*] \tag{4}$$

$$\mathcal{G}_i[p^*] \leq 0 \qquad \forall i \in [1, m] \tag{5}$$

$$\mathcal{H}_j[p^*] = 0 \forall j \in [1, n] \tag{6}$$

$$\mu_i \geq 0 \qquad \forall i \in [1, m] \tag{7}$$

$$\mu_i \mathcal{G}_i[p^*] = 0 \qquad \forall i \in [1, m] \tag{8}$$

$$\tag{9}$$

Determining the extended likelihood requires solving the problem

$$\underset{p}{\text{maximize}} \qquad -\int_{\mathcal{Y}} \mathrm{d}\boldsymbol{y}\, p(\boldsymbol{y}) \log p(\boldsymbol{y}) \tag{10}$$

$$\text{subject to} \qquad \rho(\boldsymbol{x}^*, \boldsymbol{x}) \geq \int_{\mathcal{Y}} \mathrm{d}\boldsymbol{y}\, p(\boldsymbol{y}) \log\left(\frac{p(\boldsymbol{y})}{q(\boldsymbol{y})}\right)$$

$$1 = \int_{\mathcal{Y}} \mathrm{d}\boldsymbol{y}\, p(\boldsymbol{y})$$

This problem has the following functionals.

$$\mathcal{F}[p] = -\int_{\mathcal{Y}} \mathrm{d}\boldsymbol{y}\, p(\boldsymbol{y}) \log p(\boldsymbol{y}) \tag{11}$$

$$\mathcal{H}[p] = -\rho(\boldsymbol{x}^*, \boldsymbol{x}) + \int_{\mathcal{Y}} \mathrm{d}\boldsymbol{y}\, p(\boldsymbol{y}) \log\left(\frac{p(\boldsymbol{y})}{q(\boldsymbol{y})}\right)$$

$$\mathcal{G}[p] = -1 + \int_{\mathcal{Y}} \mathrm{d}\boldsymbol{y}\, p(\boldsymbol{y})$$

The stationarity condition is then

$$\delta\mathcal{F}[p^*] = \sum_{i=1}^{m}\mu_i\delta\mathcal{G}_i[p^*] + \sum_{j=1}^{n}\lambda_j\delta\mathcal{H}_j[p^*] \tag{12}$$

$$-\int_{\mathcal{Y}}\mathrm{d}\boldsymbol{y}\,\delta p(\log p + 1) = \mu\int_{\mathcal{Y}}\mathrm{d}\boldsymbol{y}\delta p\left(\log\frac{p}{q}+1\right) + \lambda\int_{\mathcal{Y}}\mathrm{d}\boldsymbol{y}\,\delta p \tag{13}$$

$$0 = \int_{\mathcal{Y}}\mathrm{d}\boldsymbol{y}\,\delta p\left[\log p + 1 + \mu(\log\frac{p}{q}+1)+\lambda\right] \tag{14}$$

The du Bois-Reymond lemma implies the bracketed term must be zero almost everywhere.

$$0 = \log p + 1 + \mu(\log\frac{p}{q}+1)+\lambda \tag{15}$$

$$\log p = \frac{\mu}{1+\mu}\log q + \frac{\mu+\lambda+1}{\mu+1} \tag{16}$$

To satisfy the complementary slackness condition, we have either $\mu = 0$ or $\mathcal{G}[p^*] = 0$. If $\mu = 0$, stationarity implies

$$\log p = \lambda + 1 \tag{17}$$

It follows that either $p(\boldsymbol{y}) \propto 1$ or else $\mu > 0$.

If $\mu > 0$, then $\frac{\mu}{\mu+1} \in (0,1)$; moreover, the map $f_k : a \to a^k \quad \forall a \in \mathbb{R}$ is subadditive for all $k \in [0,1)$. Consequently, since $q$ is measurable, $q^{\frac{\mu}{1+\mu}}$ is measurable, and $\exists\lambda \in \mathbb{R} < \infty$ such that

$$p \propto q^{\frac{\mu}{1+\mu}} \tag{18}$$

is a well-formed probability distribution with the same support as $q$. To ensure this solution does in fact maximize the entropy, consider the second variation.

$$\mathcal{F}[p] = -\int_{\mathcal{Y}}\mathrm{d}\boldsymbol{y}\,p\log p \tag{19}$$

$$\mathcal{F}''[p] = -\int_{\mathcal{Y}}\mathrm{d}\boldsymbol{y}\,\frac{1}{p} \tag{20}$$

Since $p \geq 0$, it follows that $\mathcal{H}''[p] < 0\forall\mu \geq 0$, and thus this is a maximizing solution.

## 2 The kernel function

Consider the extended likelihood function $p$ based on $q$.

$$p(\boldsymbol{y}) \propto q(\boldsymbol{y})^{k(\boldsymbol{x},\boldsymbol{x}')} \tag{21}$$

If $q(\boldsymbol{y})$ is a known distribution in the exponential family, we may rewrite this in terms of its natural parameterization.

$$p(\boldsymbol{y}) \propto \exp\left(k(\boldsymbol{x},\boldsymbol{x}')\boldsymbol{\eta}^\top\boldsymbol{T}(\boldsymbol{y})\right) \tag{22}$$

The KL divergence between two distributions in the same exponential family can be written in terms of the log-partition function $A(\boldsymbol{\theta})$, as described by Nielsen and Nock [1].

$$\begin{aligned}D_{\mathrm{KL}}\left(p\|p_0\right) =&A(\boldsymbol{\theta}) - A\left(k(\boldsymbol{x},\boldsymbol{x}')\boldsymbol{\theta}\right)\\ &- \left(1 - k(\boldsymbol{x},\boldsymbol{x}')\right)\boldsymbol{\eta}^\top\nabla A\left(k(\boldsymbol{x},\boldsymbol{x}')\boldsymbol{\theta}\right)\end{aligned} \tag{23}$$

By construction $k(\boldsymbol{x},\boldsymbol{x}')$ will be chosen to enforce a bound on the KL divergence; that is, $D_{\mathrm{KL}}\left(p\|p_0\right) = \rho(\boldsymbol{x},\boldsymbol{x}')$. This implicitly defines a constraint on $k(\boldsymbol{x},\boldsymbol{x}')$, which must hold independent of the particular choice of $\boldsymbol{\theta}$.

$$\begin{aligned}\rho(\boldsymbol{x},\boldsymbol{x}') =&A(\boldsymbol{\theta}) - A\left(k(\boldsymbol{x},\boldsymbol{x}')\boldsymbol{\theta}\right)\\ &- \left(1 - k(\boldsymbol{x},\boldsymbol{x}')\right)\boldsymbol{\eta}^\top\nabla A\left(k(\boldsymbol{x},\boldsymbol{x}')\boldsymbol{\theta}\right)\end{aligned} \tag{24}$$

Consider the derivative of this expression with respect to $k$.

$$\frac{\partial \rho}{\partial k} = \left(k(\boldsymbol{x}, \boldsymbol{x}') - 1\right)\boldsymbol{\eta}^\top \nabla\nabla^\top A\left(k(\boldsymbol{x}, \boldsymbol{x}')\boldsymbol{\eta}\right)\boldsymbol{\eta} \tag{25}$$

Well-known properties of the log-partition function imply the Hessian of the log-partition function is the covariance of the sufficient statistic $\boldsymbol{T}(\boldsymbol{y})$, and is thus positive definite.

$$\nabla\nabla^\top A(\boldsymbol{\eta}) = \mathrm{Cov}[\boldsymbol{T}(\boldsymbol{y})] \succ \boldsymbol{0} \tag{26}$$

This implies that $\rho(\boldsymbol{x}, \boldsymbol{x}')$ is monotonically decreasing in $k(\boldsymbol{x}, \boldsymbol{x}')$ in the interval $[0, 1]$.

$$\frac{\partial \rho(\boldsymbol{x}, \boldsymbol{x}')}{\partial k(\boldsymbol{x}, \boldsymbol{x}')} < 0 \qquad \forall k(\boldsymbol{x}, \boldsymbol{x}') \in [0, 1] \tag{27}$$

Note also that we may solve for $k$ when $\rho = 0$, obtaining that $k(\boldsymbol{x}, \boldsymbol{x}') = 1$ and $\frac{\partial \rho}{\partial k} = 0$. Provided the maximum entropy distribution is not the constant distribution, this allows us to write $k(\boldsymbol{x}, \boldsymbol{x}')$ as a line integral over some curve $C$ connecting $\boldsymbol{x}$ and $\boldsymbol{x}'$.

$$\begin{aligned} k(\boldsymbol{x}, \boldsymbol{x}') &= 1 + \int_C \frac{\partial k}{\partial \rho} \nabla\rho(\boldsymbol{x}, \boldsymbol{x}')^\top \mathrm{d}\boldsymbol{x}' \\ &= 1 + \int_0^1 \frac{\partial k}{\partial \rho} \nabla\rho(\boldsymbol{x}, \boldsymbol{C}(s))^\top \frac{\mathrm{d}\boldsymbol{C}}{\mathrm{d}s} \mathrm{d}s \end{aligned} \tag{28}$$

Evaluating this expression for arbitrary $\rho(\boldsymbol{x}, \boldsymbol{x}')$ may be difficult; for many common distributions it cannot be done in terms of elementary functions. However, a solution exists, and conversely, if a function $k(\boldsymbol{x}, \boldsymbol{x}') \in [0, 1]$ is specified, we may obtain an equivalent KL divergence bound.

$$\begin{aligned} \rho(\boldsymbol{x}, \boldsymbol{x}') &= \int_C \frac{\partial \rho}{\partial k} \nabla k(\boldsymbol{x}, \boldsymbol{x}')^\top \mathrm{d}\boldsymbol{x}' \\ &= \int_0^1 \frac{\partial \rho}{\partial k} \nabla k(\boldsymbol{x}, \boldsymbol{C}(s))^\top \frac{\mathrm{d}\boldsymbol{C}}{\mathrm{d}s} \mathrm{d}s \end{aligned} \tag{29}$$

Note that the restriction on the range of $k(\cdot, \cdot)$ is important to ensure that $\rho(\cdot, \cdot)$ is positive and hence a valid bound for the KL divergence. Because the computation of the posterior distribution for a kernel process depend solely on the kernel function $k(\cdot, \cdot)$ and not on the KL bound $\rho(\cdot, \cdot)$, we may choose any suitable $k(\cdot, \cdot)$ and be satisfied that an equivalent $\rho(\cdot, \cdot)$ exists, without ever explicitly evaluating that $\rho(\cdot, \cdot)$.

## 3 A closed-form relation for the normal distribution

Suppose the base distribution is a normal distribution $\mathcal{N}(\boldsymbol{\mu}, \boldsymbol{\Sigma})$; then the extended likelihood will also be a normal distribution with the covariance scaled by $k$:

$$p = \mathcal{N}\left(\boldsymbol{\mu}, \frac{1}{k}\boldsymbol{\Sigma}\right) \tag{30}$$

The Kullback-Liebler divergence between these distributions is

$$D_{\mathrm{KL}}\left(\mathcal{N}\left(\boldsymbol{\mu}, \frac{1}{k}\boldsymbol{\Sigma}\right) \| \mathcal{N}(\boldsymbol{\mu}, \boldsymbol{\Sigma})\right) = \frac{1}{2}\left(\mathrm{tr}\left(\boldsymbol{\Sigma}^{-1}\frac{1}{k}\boldsymbol{\Sigma}\right) - d - \log\left(\frac{\det \frac{1}{k}\boldsymbol{\Sigma}}{\det \boldsymbol{\Sigma}}\right)\right) \tag{31}$$

$$= \frac{d}{2}\left(\frac{1}{k} - 1 - \log\frac{1}{k}\right) \tag{32}$$

where $d$ is the dimension of the distribution. If the divergence satisfies a bound $\rho = \rho(\boldsymbol{x}, \boldsymbol{x}')$, we have

$$\rho(\boldsymbol{x}, \boldsymbol{x}') = \frac{d}{2}\left(\frac{1}{k(\boldsymbol{x}, \boldsymbol{x}')} + \log\frac{1}{k(\boldsymbol{x}, \boldsymbol{x}')} - 1\right) \tag{33}$$

This allows us to compute the equivalent bound $\rho(\cdot, \cdot)$ for any valid kernel $k(\cdot, \cdot)$, and illustrates why it is important that $k(\cdot, \cdot) \in [0, 1]$, as kernel values outside that range result in negative or imaginary divergence bounds. The inverse relationship cannot be solved in terms of elementary functions, but has a solution in closed form in terms of the Lambert W function.

$$k(\boldsymbol{x}, \boldsymbol{x}') = -W_{-1}\left(-\exp\left(\frac{2}{d}\rho(\boldsymbol{x}, \boldsymbol{x}') + 1\right)\right) \tag{34}$$

This function is normalizable, finite, and monotonically decreasing as $\rho$ increases, just as the differential analysis above predicts.

# References

[1] F. Nielsen and R. Nock, "Entropies and cross-entropies of exponential families," in *Image Processing (ICIP), 2010 17th IEEE International Conference on*. IEEE, 2010, pp. 3621–3624.