[Reviews · NeurIPS 2014]

Submitted by Assigned_Reviewer_9

The authors present a model for latent functions in regression with which exact inference is possible if the likelihood function is chosen from an exponential family. The latent function is selected with the imposition of a smoothness constraint on a kernel function (of the covariates), which is chosen with information theoretic methods relating proposal distributions.

I think the procedure is well motivated with the intention of sacrificing the general flexibility of Gaussian processes for a very favorable complexity of inference. I think the construction of the model and model selection methods are very clean and intuitive. I appreciate the snippets of insight provided throughout Sec. 2. I am very happy that the authors included a discussion of when their modeling assumptions are and are not appropriate (in the conclusion), but I would have like to additionally have seen some specific example applications of when these modeling assumptions are appropriate.

The experiments are very convincing, though again it should be stressed that these comparisons are only meaningful in situations where your strong modeling assumptions are appropriate.
Summary: A well motivated, intuitive, and clean model derivation for a method to learn latent functions in regression models. There is good discussion of the situations in which their modeling restrictions are not prohibitive, and the experiments show excellent performance in such situations.

Submitted by Assigned_Reviewer_13

This paper presents a novel way to infer posterior distributions in Gaussian process-like distributions. The approximation is similar to Snelson, and each training sample is independent of other training samples given the single target point. The proposed idea removes the generative model on the prior in exchange for easy computations on the posterior distribution using an idea of statistical smoothness based on the KL divergence and the maximum entropy distribution. The computations on the posterior can be computed in O(N) (or possibly O(logN) in special cases) dependent on a kernel function, and the posterior result has relationships with known classical estimators. Results are fast, and very good for their computational complexity. The paper is fairly well-written.

The biggest issue with this paper is the lack of discussion of the kernel function. It is not discussed how to set it and in the experiments no details on the kernel were given in the experiments section, so it's unclear how much the results are dependent on the appropriate choice of kernel function.

Also, it should be noted in the trade-offs in the discussion that the proposed model gives independent estimates for multiple target points, where as in a GP the multiple target points are correlated.
Summary: This paper presents an interesting and probably useful approximation in GP-like model for exponential families, but needs some additional information on choosing kernels for the method and further details on experiments.

Submitted by Assigned_Reviewer_15

Brief Discussion:

The paper presents a method for generalisation of local kernel methods to the Bayesian stochastic process setting. In particular they define a smoothness condition that allows them to obtain under certain assumptions a link between a Guassian process kernel and the smoothness condition imposed on the likelihood.

Quality
The paper is well written but would benefit from more detailed discussion on the relationship between the kernel and the function rho(.,.) for smoothness. In particular some examples and discussion on whether this is always possible to achieve for any class of kernel would be beneficial. This could be done directly after Theorem 1. In addition, the results presented on the extended likelihood could be more carefully explained and discussed.

The authors should further motivate the assumption of independence which does hold in general settings - what are the settings this is reasonable in practice, what are the implications of this assumption on the inference otherwise in practical examples proposed. Explain further how it relates to the other methods cited in this section on sparsification - what exactly is the relationships perhaps make a small illustrative example of the link. In addition the discussion on the equivalence of the interaction between input variables either in the likelihood function or in the prior is interesting and should be expanded. In practical settings which would you choose in general to model the interaction in the likelihood or the prior, comment on this remark, and is this equivalence always true - under what conditions? please comment.

Clarity - the paper is generally well written and concise

Originality - the paper is interesting though the contributions are incremental in nature

Signficance - the paper is interesting for the audience.

Summary: The paper is well written and interesting though it lacks sufficient description of the relationship between a covariance kernel k(.,.) and the smoothness metric on the extended likelihood to be useful. In addition, it should be more carefully explained and detailed with examples of the conversion / equivalence discussed.

Author Feedback
Author rebuttal: We thank the reviewers for their helpful commentary. There is consensus among reviewers that we did not supply enough detail regarding the kernel function in theory or in practice; if accepted we will ensure this concern is addressed.

Reviewer 13 notes that we did not discuss how to choose a kernel function or what choice we made in our experiments. We agree we should not have omitted these details; in all our experiments, we used the squared exponential kernel exp(-(x-x')^2/s^2), where s is a scalar parameter describing smoothness. We found this kernel effective across a variety of problems. If accepted, we will amend the paper to include both details of the kernel we used, as well as some guidance on choosing kernels for other problems.

Reviewer 15 questioned whether a smoothness function rho(.,.) exists for any class of kernel k(.,.). We appreciate the feedback that the paper was not clear on this point; in fact a smoothness function rho(.,.) exists for any kernel function which is bounded above by 1, and with the property that k(x,x)=1. The corresponding distance function can be given explicitly as the solution to a differential equation for any distribution in the exponential family. We discuss this briefly in our supporting material, but agree that the provided detail is insufficient; we will add an explicit example mapping from k to rho.

In addition, reviewer 15 found the discussion on the extended likelihood unclear. We will provide more explanation and discussion in the final revision; however, we would be grateful for any additional clarification from the reviewer as to whether the section as a whole requires additional explanation or if there are specific points that are unclear.